# Experimental Characterization of the Torsional Damping in CFRP Disks by Impact Hammer Modal Testing

**DOI:** 10.3390/polym12020493

**Published:** 2020-02-24

**Authors:** Francesco Cosco, Giuseppe Serratore, Francesco Gagliardi, Luigino Filice, Domenico Mundo

**Affiliations:** Department of Mechanical, Energy and Management Engineering, University of Calabria, Cubo 45C, 87036 Rende, Italy; francesco.cosco@unical.it (F.C.); giuseppe.serratore@unical.it (G.S.); luigino.filice@unical.it (L.F.); domenico.mundo@unical.it (D.M.)

**Keywords:** vibration test, impact hammer test, torsional mode, polymer-based composite materials, energy loss

## Abstract

Composite materials are widely used for their peculiar combination of excellent structural, mechanical, and damping properties. This work presents an experimental study on the dissipation properties of disk-shaped composite specimens exploiting vibration tests. Two different polymer matrix composites with the same number of identical laminae, but characterized by different stacking sequences, namely unidirectional and quasi-isotropic configurations, have been evaluated. An ad-hoc steel structure was designed and developed to reproduce an in-plane torsional excitation on the specimen. The main idea of the proposed approach relies on deriving the damping properties of the disks by focusing on the modal damping of the overall vibrating structure and, in particular, using just the first in-plane torsional deformation mode. Experimental torsional damping evaluations were conducted by performing vibrational hammer excitation on the presented setup. Two methods were proposed and compared, both relying on a single-degree-of-freedom (SDOF) approximation of the measured frequency response function (FRF).

## 1. Introduction

In the context of ever-increasing performance demands, and pushed by the ultra-rapid technological progress of the last century, composite materials have been adopted in several industrial domains. This is thanks to their intrinsic ability to offer a more compact, lightweight alternative, with the same or similar strength and stiffness, and even better dynamic performance. Substantial research efforts have been dedicated to further develop these innovative materials, such as reinforced epoxy nanocomposites [1], nitinol (NiTi) shape-memory-alloy-based composites [2], fiber-reinforced polymer composites [3], and carbon fiber-reinforced polymer (CFRP) composite materials [4,5,6,7]. CFRPs represent a particularly attractive choice, the versatility of which has attracted attention not only in the aeronautic [8] and automotive [9] sectors, but also in many other engineering applications, such as the production of advanced machining tools [4,5,6], reinforcement implants for constructions [10], and spacecraft applications [11].

CFRP success relies not only on combining structural strength and stiffness with being light weight, but also on exhibiting superior damping characteristics as a result of the polymeric matrix, which makes them valuable allies for mitigating noise and vibration problems [9]. Indeed, the damping capacity of a material is evaluated by measuring the energy dissipated during mechanical vibrations [12]. As opposed to traditional structural materials (such as steel and lightweight alloys), polymer-based composite materials are able to dissipate more energy, not only because of the presence of the polymeric fraction, but also thanks to the positive interaction between the fiber and matrix [13]. Aggogeri et al. [4] evaluated the adoption of a CFRP prototype for the *Z*-axis ram of a commercial milling machine so as to reduce the effects of the undesired vibrations and to increase the workpiece quality. The experimental results showed that the CFRP structure guaranteed a high stiffness with a weight reduced by 48.5%, and an almost three-fold increase in damping compared with steel alternatives. Xie et al. [10] studied the mechanical properties and characteristics of CFRP stay cables in a cable-stayed bridge. Their finding confirmed that the use of CFRP instead of steel in making cables was instrumental in preventing the cable-bridge vibration coupling, and resulted in a better attenuation of the parametric and wind fluctuating vibrations, with the vibration amplitudes of the CFRP cables halved with respect to the steel cable solution.

The effective use of these innovative materials, including CFRP composites, within engineering design daily practices demands not only the proposition of new material modelling strategies [6,11], but also the definition of dedicated testing procedures, targeted to characterize the mechanical properties [12,14], as well as inspection procedures to assess fatigue, detect defects, and eventually identify the related failure mode [8]. In this scenario, modelling and assessing the damping phenomena in composite materials have also received vast attention, as documented in the literature [13,15]. Specifically, the experimental assessment of torsional damping properties has been documented for different materials or specimens [16,17,18], but, so far, minimal attention has been dedicated to the torsional dynamic characterization of flat disks, which is crucial for the design of CFRP inserts aiming at mitigating vibrations in rotor dynamics applications.

Besides the well consolidated approach based on exploiting the dynamic mechanical analysis (DMA) [19,20,21], the feasibility of assessing the damping properties through vibration analysis has been recently demonstrated [22,23,24]. V. Anes et al. [22] investigated the torsional and axial damping properties of a bar-extruded AZ31B-F magnesium alloy, considering both hysteretic and viscous damping with stress–strain-controlled tests and free vibration analyses. Rueppel et al. [23] studied the damping performance of the CFRP laminated plates in different laminate angles with logarithmic decrement, and compared the test results obtained by three damping methods.

Recently, Cosco et al. proposed a dedicated testing setup [20], relying on the DMA approach for the experimental assessment of the stiffness and energy dissipation properties of disk-shaped polymer-based composite specimens by in-plane torsion testing. This paper complements the experimental findings presented in reference [22], with an improved apparatus and a novel approach to evaluate the torsional damping of CFRP specimens using vibration analysis. The dedicated apparatus was designed to hold the disk-shaped samples while maintaining the first in-plane torsional mode of the disk specimen in a frequency range small enough to be excited uniformly by impact hammer tests. The proposed methodology relies on extracting the damping properties from one frequency response function, obtained by correlating the input hammer force with the vibrations of the structure on a nearby location.

The outline of the paper is as follows: Section 2 describes the developed experimental apparatus along with a summary of the mathematical background of a single-degree-of-freedom (SDOF) frequency response function (FRF) modal model approximation, which represents the common foundation for the two methods adopted for extracting the damping parameters, namely the half-power bandwidth and a more elaborate least-squares fitting approach. The proposed SDOF FRF fitting offers the benefits of using more points for the estimation, thus increasing its robustness against possible sources of noise affecting the measured FRF. In Section 3, both of the proposed methods are employed to assess and compare the damping properties of one steel disk and two CFRP disks (one produced to mimic quasi-isotropic behavior, and the other with a unidirectional orientation of the plies). Concluding remarks are provided in Section 4, along with an outlook on the future developments of the presented research work.

## 2. Materials and Methods

### 2.1. Materials

The CFRP composite disks analyzed in this work were fabricated using carbon fiber dispersed in about 37 wt % of an epoxy matrix. Two different laminate layups were considered for the CFRP specimens. The first was made with 24 plies oriented along the same 0° direction (unidirectional configuration (UD)), while the second one was composed of the same number of plies with a symmetric (both geometric and material properties symmetric about the middle plane) and a quasi-isotropic (QI) configuration. Specifically, the in-plane isotropy was obtained with a pattern of four plies (0, 45, 90, −45), repeated six times through the disk thickness [25]. The reference disk specimen and the test-rig components were fabricated with regular steel. The mechanical properties of all of the materials are listed in Table 1.

### 2.2. Impact Hammer Modal Testing Equipment

The experimental setup, derived from the one used in the literature [20] for DMA-based tests, is shown in Figure 1, and consists of a steel structure designed to enable the in-plane vibration excitation of the specimen by impact hammer modal testing. The proposed experimental methods rely on the following equipment:a SCADAS mobile system;a laptop with Simcenter Testlab Impact Testing software;an impact hammer;a set of uniaxial accelerometers;the holding steel structure;and the sample disks.

Impact hammer modal testing is a commonly used method in the field of vibration analysis, enabling the measurement of FRFs of a vibrating structure. Each FRF represents the spectrum of the vibration of a single point on the structure, in a particular direction, and in response to a unit force applied in some other location of the structure.

### 2.3. Layout of the Specimens and Testing Procedure

Three different disk specimens, one made of steel, and two CFRP disks with different laminate layups (one with a quasi-isotropic (QI) configuration, and the other with a unidirectional (UD) configuration) were tested. The latter of the CFRP disks was tested in two different positions, UD-0° and UD-90°, with the layups oriented to be parallel and orthogonal, respectively, to the moment arm used for hammer excitation, as depicted at the bottom of Figure 2. Summarizing, we planned four different testing configurations, namely: Steel, QI, UD-0°, and UD-90°.

The composite disk specimens had an outer diameter of 195 mm, a thickness of 4 mm, and a 20 mm diameter central hole in order to ensure, together with a centering pin, a precise position in the proposed set-up. Instead, the steel specimen was manufactured with a thickness of 1.15 mm in order to obtain approximatively the same torsional stiffness as the QI composite specimen.

As depicted in Figure 2, each disk specimen was fixed by means of a double flanged connection relying on friction: the fixed clamp blocking the outer portion of the disk on the fixed lower arm, while the inner clamp fixed the moment arm at the center of the disk. The impact hammer and the accelerometer were used to measure the induced force and the resulting vibration at the tip of the moment arm. The testing protocol used for measuring the FRF was provided by the Impact Testing module of the Siemens Simcenter TestLab software (Siemens, Leuven, Belgium) running on the laptop used to steer the data acquisition system (SCADAS Mobile). For each testing configuration, a set of five different impacts was recorded with a sampling frequency of 512 Hz and a target resolution frequency of 0.25 Hz. The resulting average FRF was saved for further post-processing in MATLAB.

### 2.4. Torsional Damping Assesment Using Vibration Analysis

The idea behind this work is to evaluate the torsional damping properties of the composite disk specimen by isolating the motion components equivalent to an in-plane torsion condition.

Figure 3a depicts a schematic representation of the testing setup, where a beam is clamped on a compliant joint. The beam represents the moment arm, while the joint represents the disk specimen. An equivalent damped SDOF system for describing the overall motion of the other free-to-vibrate end of the arm is schematized in Figure 3b. In general, the SDOF system representation can be misleading, owing to the multiple sources of motions, all contributing to the vertical displacement of the free-end of the moment arm, which must be considered in the real testing setup, such as the bending deformation of the beam, the eventual displacement of the fixed lower arm, and the torsional deformation of the disk itself, which will be reflected as a rigid rotation of the beam around the compliance joint. Therefore, the adoption of an SDOF approximation is valid only if two strong requirements are verified: a clear distinction among the all-possible causes of motion (requirement 1) and the presence of a limited frequency band, where effects other than the in-plane deformation are negligible (requirement 2).

The modal analysis theory [26] offers the proper tools to understand and assess the limits of such equivalent systems. Indeed, the parametric modal model, usually expressed as follows [27]:(1)$$H_{jk}\left(\omega \right)\approx \overset{N_{m}}{\underset{r=1}{\sum }}\left(\frac{\psi_{jr}\;L_{rk}}{j\omega -\lambda_{r}\;}+\frac{\psi_{jr}^{*}L_{rk}^{*}}{j\omega -\lambda_{r}^{*}}\right)+UR_{jk}-\frac{LR_{jk}}{\omega^{2}},$$
enables the expression of a matrix of the measured FRFs, defining all of the transfer functions, $H_{jk}\left(\omega \right)$, between the *k*th input force and *j*th displacement response. Each FRF is expressed as the superposition of the first $N_{m}$ eigenmodes, each of which is characterized by a pair of complex conjugate poles, $\lambda_{r}=\sigma_{r}+j\omega_{r}\;$ and $\lambda_{r}^{*}=\sigma_{r}-j\omega_{r}$, where $\omega_{r}$ and $\sigma_{r}$ are the damped natural frequency and the damping factor, respectively; $\psi_{jr}$ is related to the eigenshape; and $L_{rk}$ is the participation factor, which depends not only on the corresponding eigenshape, but also on the efficiency by which each mode has been excited from the corresponding excitation degree of freedom [26]. Given the desired frequency range, a very compact yet effective modal model is built, comprising only those eigenmodes with a natural frequency falling in the desired range, whereas the lower- and upper-residual terms, LR and UR, are instrumental to model the out-of-band modes effects within the considered frequency range. Such models can also be seen as a superposition of several SDOF systems, which is usually referred to as a multiple degrees of freedom (MDOF) system.

It is worth noting that, in the case of using just one input force and one response, the modal model equation collapses to a scalar function:(2)H(ω)≈∑r=1Nm(1 jω−λr Qi+1jω−λr∗Qi∗)+UR−LRω2 with: λr, Qr, UR, LR∈ℂ

Equation (2) maintains the form of an MDOF, where we find the residue $Q_{r}$, instead of the eigenshapes and the participation factor terms [26]. Figure 4 depicts a schematic representation highlighting the above-mentioned superposition of the different modal contributions on a measured FRF. It is worth noting that, in the proximity of a resonance peak, the contributions of all of the modes, other than the resonant one, tend to vanish.

The adoption of Equation (2) satisfies the above-mentioned requirements, also offering a mathematical foundation to assess the limit of applicability of the SDOF approximation. Therefore, to fulfill the scope of this paper, it suffices to find the smallest eigenmode satisfying the required in-plane torsion conditions, in addition to being well decoupled from the other modes. The existence of such a mode was confirmed by means of a modal analysis, as also reported at the beginning of the results section. The remainder of this section describes two alternative methods, namely the half-power bandwidth and the SDOF-FRF fitting, used to compute the damping ratio from one measured FRF.

#### 2.4.1. Estimating Modal Damping Ratio Using the Half-Power Bandwidth Method

The half-power bandwidth method is commonly used for estimating the modal damping related to a given eigenmode. The method consists of finding the magnitude of the resonance peak within the measured FRF corresponding to the chosen mode, and then evaluating the locations of the upper- and lower-bound frequencies, $\omega_{lo}$ and $\omega_{up}$, corresponding to a 3 Db decrease from the peak amplitude of the damped resonant frequency, ω_r_. The modal damping ratio is computed as follows:(3)$$\zeta_{r}=\frac{\omega_{up}-\omega_{\mathrm{lo}}}{2\omega_{r}}.$$

Relying on a limited amount of information, as only three points of the measured FRF are used, this method is very fast to compute. However, it may exhibit precision degradation in the presence of noise in the FRF.

#### 2.4.2. Estimating Modal Damping Ratio Using the SDOF FRF Model

On the premises that, around a single peak, only one modal contribution is dominant, Equation (2) can be further simplified, resulting in the parametric form of an SDOF FRF model approximation:(4)$$H\left(\omega \right)\approx \frac{1}{M\left(\omega_{n}{}^{2}-\omega^{2}+j\;2\;\omega_{n}\;\omega \;\zeta \right)}+UR-\frac{LR}{\omega^{2}}$$
where M is the modal mass, $\omega_{n}$ is the eigenfrequency, and ζ is the damping ratio. The estimation of all of the parameters, including the damping ratio, was accomplished using the simplex search method [28] and a nonlinear fitting approach implemented in MATLAB.

## 3. Results

### 3.1. Identification of the First in-Plane Torsional Eigenmode by Modal Analysis

As a preliminary investigation, a set of FRFs was collected by means of impact hammer modal testing using a set of seven accelerometers dislocated along the structure, with the steel disk specimen mounted. A modal analysis was executed using the dedicated module within the LMS TestLab software, the goal being to fit the parametric modal model of Equation (2) on top of the measured FRFs.

Figure 5 depicts the measured FRF corresponding to the moment arm location. The obtained modal model is composed of five modes, as detailed in Table 2. After revising the corresponding eigenshapes, we concluded that the last mode in the analyzed frequency band satisfied the condition of in-plane torsion, i.e., showing a rotation of the moment arm around the center of the disk. However, it also showed a minor effect of vertical bouncing with respect to the lower structure, because of the deviation of the real boundary conditions from an ideal clamping. Figure 6 depicts the eigenshape of the mentioned torsional mode. An animation video is also attached as additional material.

### 3.2. SDOF-FRF-Based Modal Damping Characterization of the First in-Plane Torsional Eigenmode

This section summarizes the damping estimation results. Figure 7 depicts the SDOF fitting results obtained by the method described in Section 2.4.2. The good agreement between the experimental data and the fitted model confirms the validity of the proposed methodology and its underlaying assumptions.

The UD-0° specimen disk exhibited the highest modal damping ratio, and thus the highest torsional vibration attenuation. However, it is noteworthy that the estimated damping was greatly influenced by the position of the fibers. Indeed, as already discussed in the previous section, the moment arm was not only rotating around the disk, but also bouncing in a vertical direction. This motion was highly sensitive to the fiber orientation. Specifically, in the UD-0° configuration, the polymeric matrix was subjected to a relatively larger deformation owing to the fibers being orthogonal to the bouncing motion. Meanwhile, in the UD-90°, the fibers reacted along the bouncing direction, causing a drastic increase in stiffness, thus reducing the amount of vertical deformation and, consequently, the amount of dissipated energy. This fiber action resulted in both a relative increase in the modal stiffness, as shown by the eigenfrequency shift to higher values, and a drastic reduction of the damping capabilities.

In order to further validate the proposed approach, the damping ratios were also calculated using the half-power bandwidth method, according to Equation (3). A comparison of the damping ratio values obtained using both calculation methods is reported in Table 3.

As expected, the modal damping ratios estimated by the two methods were very close to one another in each investigated case, validating the robustness of the proposed methodologies. For the steel case, the extracted values (0.0109 and 0.0110) corresponded to the one computed by means of the modal analysis identification (0.0111), thus confirming the validity of the SDOF approximation.

Focusing on the properties of the investigated engineering materials, the reported results show that a QI composite disk, made with approximatively the same torsional stiffness of a steel component, was characterized by a relative increase in the modal damping ratio of about 40%. This evidence is ascribable to the effect of the viscoelasticity of the polymeric matrix, which is the main contributor to the improvement of the damping property of the disk. This was further validated by looking at the UD-0° case, where the matrix was more stressed and the damping properties of the disk were nearly doubled, with a relative increase of more than 80% compared with the metallic case. Therefore, by working on the constituents, arrangement, and boundary conditions, fiber reinforced polymers can be tailored to customize the damping properties of a component, resulting in promising materials for designing and manufacturing emerging solutions in lightweight applications.

## 4. Conclusions

In this paper, a novel approach to evaluate the damping contribution of CFRP disks subject to in-plane torsion was presented. The conceived testing approach aimed at a comparative characterization of two different composite specimens, starting from the response of the proposed set-up with a steel specimen. The choice of using fiber-reinforced polymers to test the quality of the proposed approach can be ascribed to the relevant damping properties of these materials, which are highly customizable, owing to the fiber arrangements. Consequently, a proper damping evaluation is strongly required for a more effective component design.

Experimental impact hammer tests were performed by changing the specimens and the orientation of the layups. FRFs were acquired and further analyzed using the half-power bandwidth method and an SDOF fitting scheme. The results obtained by the two methods showed a good agreement, confirming the validity of the proposed approach in extracting the modal parameter information. For the chosen cases, the experimental results showed that a significant effect was noticeable in terms of the damping contribution with the composite specimens, in comparison with the steel specimen.

The proposed experimental set-up and testing procedure could also be used to detect failure and delamination [29], or to test the interfacial bond behavior between CFRP and other materials in multi-material structures [30], as the energy dissipation capacity of composites is affected by structural health and boundary conditions.

Further investigations will focus on increasing the robustness of the proposed approach, additionally considering the MDOF model and the possibility to use shaker excitation instead of hammer excitation. This would allow for the excitement of the entire equipment in a more controllable and repeatable way, and to consider different excitation amplitudes, so that possible nonlinearities in the system response can be analyzed.

## Figures and Tables

**Figure 1 polymers-12-00493-f001:**
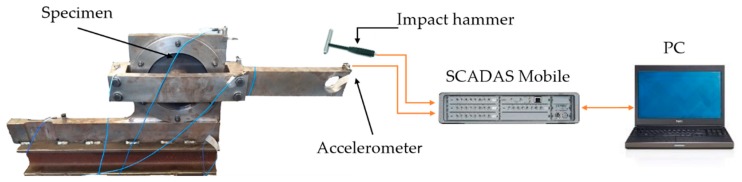
The experimental test set-up for impact hammer modal testing.

**Figure 2 polymers-12-00493-f002:**
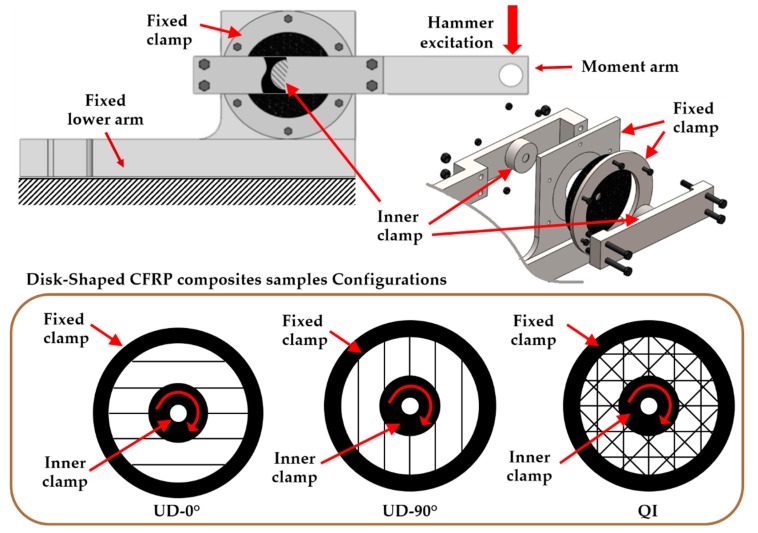
Drawing of the designed frame setup and disk-shaped sample configurations. CFRP—carbon fiber-reinforced polymer.

**Figure 3 polymers-12-00493-f003:**
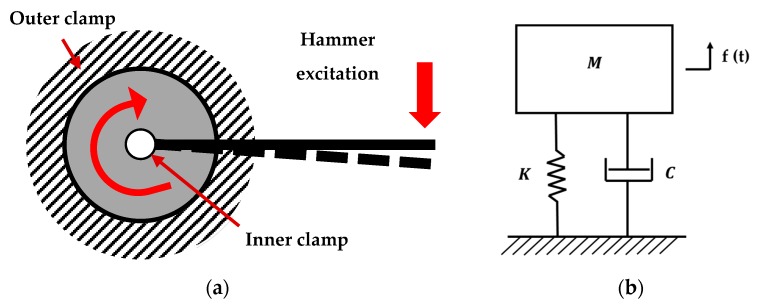
(**a**) Schematic diagrams of the testing setup: the moment arm behaves as a beam clamped on a rotational compliance; (**b**) an equivalent damped single degree of freedom system, where *M*, *K*, and *C*, represent the lumped mass, stiffness, and damping, respectively, of the depicted system.

**Figure 4 polymers-12-00493-f004:**
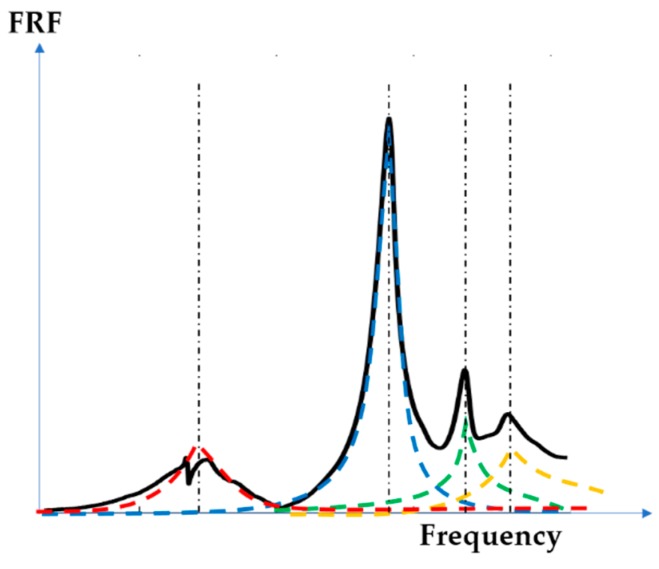
Modal model as the superposition of different single-degree-of-freedom (SDOF) contributions. FRF—frequency response function.

**Figure 5 polymers-12-00493-f005:**
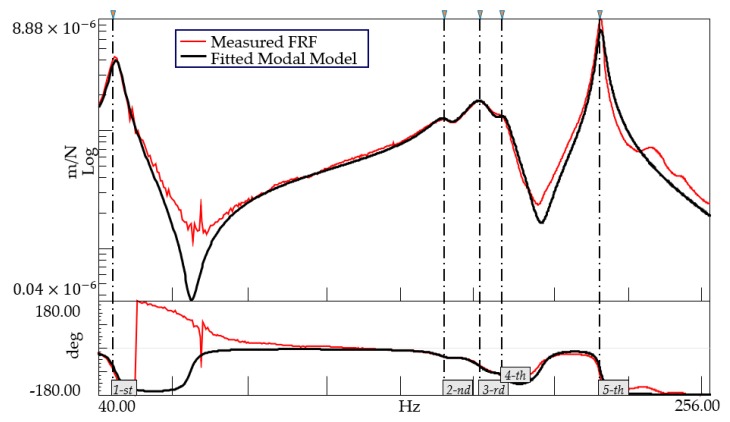
Experimental modal analysis of the set-up with the steel disk. The fitted modal model was obtained using five poles, for which the frequencies and damping ratios are reported in Table 2.

**Figure 6 polymers-12-00493-f006:**
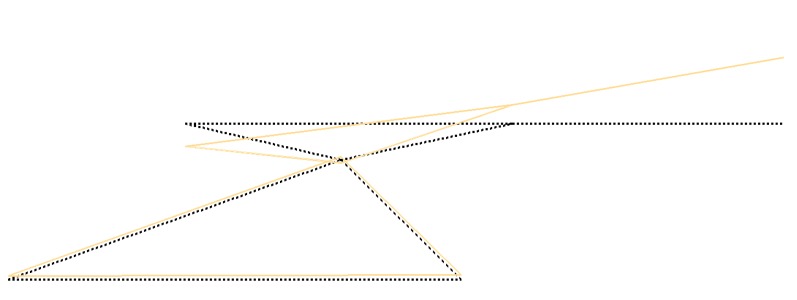
Modal shape of the in-plane torsional mode for the steel specimen case.

**Figure 7 polymers-12-00493-f007:**
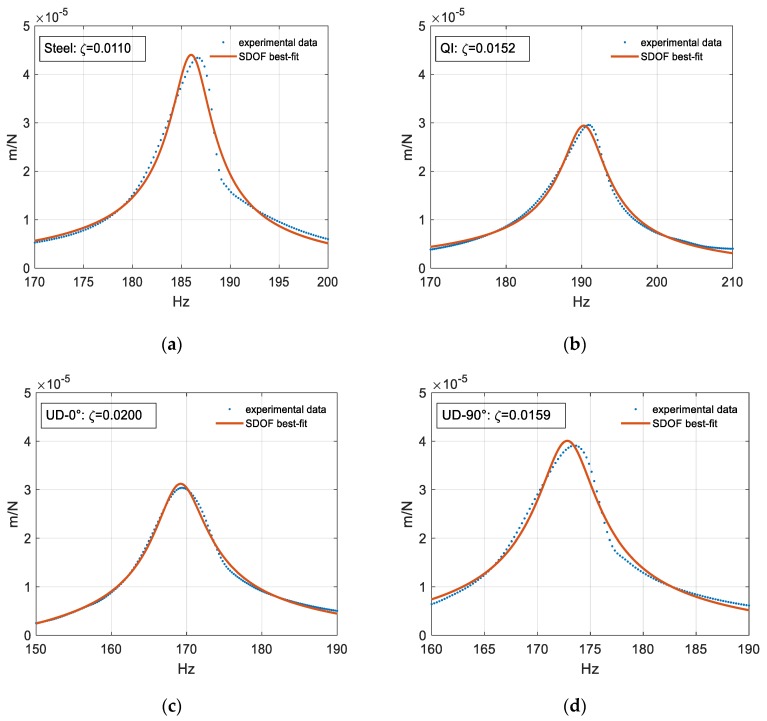
Experimental data and best-fit SDOF model for the set-up with (**a**) the steel specimen, (**b**) the quasi-isotropic (QI) specimen, (**c**) the unidirectional configuration (UD)-90° specimen, and (**d**) the UD-0° specimen.

**Table 1 polymers-12-00493-t001:** Material properties of the composite specimens and steel equipment.

Property	Specimen		Equipment	
	Fiber	Matrix	Structure	Bolts
Longitudinal Modulus (GPa)	436	2.7	220	210
Transverse Modulus (GPa)	12.35–24.78	2.7	220	210
Poisson’s ratio	0.41	0.85	0.29	0.29
Density (g/cm^3^)	1.84	1.2	7.8	7.8

**Table 2 polymers-12-00493-t002:** Eigenfrequencies and damping ratio values of the fitted modal model for the set-up with the steel disk specimen.

Modes	Frequency (Hz)	Damping Ratio
1st	41.78	0.020
2nd	114.22	0.0346
3rd	127.58	0.0511
4th	136.264	0.0259
5th	183.574	0.0111

**Table 3 polymers-12-00493-t003:** Comparison of the modal damping ratios of the overall set-up calculated using the half-power bandwidth method and the SDOF fitting.

Specimen	Half Power Bandwidth (ζ)	Increase w.r.t. Steel Specimen (%)	SDOF Fitting (ζ)	Increase w.r.t. Steel Specimen (%)
Steel	0.0109	N/A	0.0110	N/A
QI	0.0151	38.5	0.0152	38.2
UD-0°	0.0197	80.7	0.0200	81.8
UD-90°	0.0157	44.0	0.0159	44.5

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
