# Peer review of "Experimental Characterization of the Torsional Damping in CFRP Disks by Impact Hammer Modal Testing"

_polymers, 2020, doi:10.3390/polym12020493_

Round 1
Reviewer 1 Report
The present paper presents a very applicative study of polymers. The paper may be published in the present form, since it is of high quality.
Author Response
We thank the reviewer to his/her considerations on our research paper
Reviewer 2 Report
Dear Authors.
Your study is quite interesting and concerns important problem – mechanical damping in CFRP laminates, the materials applied for very responsible constructions. You determined natural frequencies (eigenfrequencies) of the CFRP laminate at specific clamping and exciting conditions. It is original set of results which may be usable for designers and constructors. I think the problem is well presented and approached to be resolved in original way. However, I found several deficiencies which should be improved before potential publication.
Abstract: explain the abbreviations "DOF" and "FRF".
Page 3, Materials and Methods: you should provide a clearer description and some clear designations of specimens tested. They should be consequently used in further text. Now it is not clear enough.
Page 4, Fig.2: give some precise picture and description concerning fixing the specimen. Especially depict clearly the "inner clamp".
Page 4, Fig.3b: Explain the designation M, K and C in a caption.
General remark 1: A term "impact testing" is usualy reserved for the impact resistance tests – e.g. Charpy's, Izod, drop weight tests. I suggest you to change the term “impact testing” to “hammer excitation tests" or other, more proper for this type of tests. You should do the change in title and in whole text.
General remark 2: The study practically focuses on methodology. You should explain the observed effects from point of view of materials engineering. Please, add some paragraph concerning this problem to the section 3 (the remarks already presented in the text are too poor). And – add some "material-concerning" paragraph in conclusions as well.
Sincerely
Author Response
On behalf of all the authors, I thank the reviewer for his/her valuable remarks. The answers to each of the reported comments are reported in the attached file
